# Utilization of Aminoguanidine Prevents Cytotoxic Effects of Semen

**DOI:** 10.3390/ijms23158563

**Published:** 2022-08-02

**Authors:** Mirja Harms, Pascal von Maltitz, Rüdiger Groß, Benjamin Mayer, Miriam Deniz, Janis Müller, Jan Münch

**Affiliations:** 1Institute of Molecular Virology, University Ulm Medical Center, 89081 Ulm, Germany; mirja.harms@uni-ulm.de (M.H.); pascal.von-maltitz@uni-ulm.de (P.v.M.); ruediger.gross@uni-ulm.de (R.G.); janismueller@uni-marburg.de (J.M.); 2Institute for Epidemiology and Medical Biometry, Ulm University, 89075 Ulm, Germany; benjamin.mayer@uni-ulm.de; 3Department of Gynecology and Obstetrics, Ulm University Hospital, 89075 Ulm, Germany; miriam.deniz@uniklinik-ulm.de; 4Institute of Virology, Philipps University of Marburg, 35043 Marburg, Germany

**Keywords:** seminal fluid, cytotoxicity, polyamines, spermine

## Abstract

Studies of human semen in cell or tissue culture are hampered by the high cytotoxic activity of this body fluid. The components responsible for the cell damaging activity of semen are amine oxidases, which convert abundant polyamines, such as spermine or spermidine in seminal plasma into toxic intermediates. Amine oxidases are naturally present at low concentrations in seminal plasma and at high concentrations in fetal calf serum, a commonly used cell culture supplement. Here, we show that, in the presence of fetal calf serum, seminal plasma, as well as the polyamines spermine and spermidine, are highly cytotoxic to immortalized cells, primary blood mononuclear cells, and vaginal tissue. Thus, experiments investigating the effect of polyamines and seminal plasma on cellular functions should be performed with great caution, considering the confounding cytotoxic effects. The addition of the amine oxidase inhibitor aminoguanidine to fetal calf serum and/or the utilization of serum-free medium greatly reduced this serum-induced cytotoxicity of polyamines and seminal plasma in cell lines, primary cells, and tissues and, thus, should be implemented in all future studies analyzing the role of polyamines and semen on cellular functions.

## 1. Introduction

Human semen (SE) is a complex body fluid consisting of a cellular fraction of mainly spermatozoa, and a cell-free fraction, the seminal plasma (SP). An average human ejaculate has a volume of 2–6 mL, consists of up to 10% spermatozoa and 90% SP, and has a pH in the range of 7.2–8.0 [1]. Seminal plasma originates from the accessory organs of the male reproductive tract, and its main function is to provide a protective and feeding environment for spermatozoa and to counteract the acid vaginal environment during fertilization [2]. Studies of SE and SP are complicated by intra- and inter-donor variations, as well as time-, age-, and diet-related changes in the composition [3,4,5]. Moreover, fresh gel-like ejaculates undergo a rapid 15–30 min protease-mediated liquefaction process which results in the liberation of motile spermatozoa. Therefore, studies with SE or SP are challenging and require well-defined standard operational procedures of ejaculate collection, sampling, and storage, and should include the examination of many individual, as well as pooled samples. 

Studies investigating the effect of SE on cellular functions are further hampered due to the long-known but poorly studied cytotoxic activity of this body fluid [6,7,8,9,10,11]. Even 10- to 100-fold dilutions of SE or SP can induce cytotoxic effects in mammalian cell cultures [12,13,14], which precludes or complicates the analysis of physiologically relevant concentrations. Experiments with SP, therefore, require the establishment of experimental conditions that, on the one hand exclude cell-damaging effects of the body fluid, e.g., through dilutions or limited incubation times but, on the other hand, still allow the determination of the effect of SP on cellular parameters. Examples of such experiments are studies on the influence of SP on the infectivity of sexually transmitted viruses, such as HIV-1, or on the anti-inflammatory properties of SP in the presence of immune cells [6,12,13,14,15].

The components which are primarily responsible for the cytotoxic effect of semen are the polyamines spermidine and spermine [11,16,17,18]. These small organic molecules are present in SP at high levels, with spermidine reaching concentrations of up to 0.6 mM and spermine of 0.25 to 14 mM [19,20]. Both polyamines carry positive charges that mediate strong interactions with negatively charged molecules, such as phosphate ions, nucleic acids, and phospholipids. Spermine and spermidine are oxidized by amine oxidases in SP and fetal calf serum (FCS), leading to the formation of aldehyde intermediates, acrolein, and ammonia [21,22,23]. The formation of acrolein represents the most important toxic intermediate [18,24], as this molecule interacts strongly with deoxyguanosine in DNA, triggering base substitutions, and with nucleophilic sites in proteins (e.g., cysteine, lysine, arginine, and histidine), leading to cross-linking, changes in signaling pathways and enzymatic activities [24], and ultimately to cell death. It is common for FCS to be used as a supplement for growth media in eukaryotic cell culture as it naturally contains amine and diamine oxidases [23]. Thus, amine oxidases present in FCS-supplemented cell culture produce toxic acrolein from polyamines, explaining the high cytotoxicity of SP [21].

Consequently, blocking the activity of these enzymes that lead to polyamine oxidation and subsequent deamination should prevent acrolein production and may allow researchers to perform cell culture experiments in the presence of physiologically relevant concentrations of polyamines and/or semen. Aminoguanidine (AG), an amino derivate of guanidine, was shown to strongly inhibit Cu^2+^-dependent amine oxidases, including diamine and serum amine oxidases [23,25,26,27]. Indeed, AG has a high affinity for diamine oxidases (apparent dissociation constant, K’1 = 0.7 nM) [28] most probably due to stabilization by conformational changes or the formation of a Schiff base with the catalytic center of the enzyme [28]. As such, AG has safely been used as a cell culture supplement to prevent the cytotoxic effects of spermine and spermidine [29,30]. Thus, we here evaluated whether the addition of AG may also reduce or prevent the cytotoxic effects of SP in cell culture. 

## 2. Results

We first verified the cytotoxic activity of SE and SP. For this, pooled SE or SP derived from 50 donors was titrated on TZM-bl cells, a HeLa cell derivative widely used in AIDS research that is engineered to express the HIV-1 receptors CD4 and CCR5 [31]. TZM-bl cells were seeded at a low cell density and treated with increasing SE or SP concentrations in DMEM supplemented with 10% (*v*/*v*) heat-inactivated FCS. After 2 days of incubation at 37 °C, cell viability was determined using a colorimetric MTT assay (Figure 1a) and light microscopy (data not shown). Microscopic evaluation revealed that cells treated with SE as well as SP concentrations as low as ~1% already exhibited a spherical shape, which became more evident with higher concentrations of the body fluids. At concentrations of 10% and higher, the cells detached completely from the bottom of the cell culture plate. The MTT assay measures the conversion of a soluble MTT salt into a purple-colored formazan product in viable cells [32]. The MTT assay confirmed these observations and showed a concentration-dependent decrease in viable cells (Figure 1a), as previously shown [29,30]. Furthermore, SE or SP concentrations > 1% led to an almost complete loss of viable cells, and concentrations as low as 0.3% already resulted in a 20% reduction in viability (Figure 1a). Thus, SE as well as SP are highly cytotoxic under standard cell culture conditions involving utilization of 10% FCS, confirming previous results of ourselves and others [6,7,8,9,10,11,12,13,14,16,17,18].

We subsequently determined the cytotoxic activity of synthetic spermine and spermidine using the same experimental setup as described above. The MTT assays performed after 2 days showed that, in the presence of 10% FCS, concentrations of ≥10 µM of both polyamines were strongly cytotoxic for TZM-bl cells (Figure 1b), confirming previous data obtained with several cell lines, such as A549 lung adenocarcinoma and HCT116 colon adenocarcinoma cells [29,30], human fibroblasts [23], rat neurons [33], or murine lymphocytes [11]. To quantify the polyamine concentrations in the pooled SP sample, liquid chromatography–tandem mass spectrometry was applied [34], revealing concentrations of ~3 mM spermine, 155 µM spermidine, 80 µM putrescine, and 255 µM L-ornithine (Figure A1), in line with published data [19]. Thus, the spermine concentrations in SP are in a range that can explain the cytotoxicity of SP, and suggests that spermine is the major contributor to the cell-damaging activity in semen. 

To further investigate the role of serum in the observed effects, we titrated FCS on TZM-bl cells supplemented with PBS, 100 µM spermine (Figure 1c), or 10% SP (Figure 1d), and determined cell viability 2 days later. As expected, FCS alone had no effect on cell viability (Figure 1c). Similarly, in the absence of FCS, spermine did not result in measurable cytotoxic activity (Figure 1c). However, the addition of only 1% FCS to cells exposed to spermine resulted in complete cell death (Figure 1c). Similar results were obtained for 10% SP, which already caused 50% cell death in the absence of FCS, probably due to oxidases produced by the cells or naturally present in SP (Figure 1d). These results suggest that spermine toxicity is dependent on the presence of FCS, and that serum-free conditions may restore cell viability in the presence of otherwise toxic spermine concentrations. 

We then analyzed the cytotoxicity of spermidine and spermine under serum-free cell culture conditions. For this, cells were either supplemented with a chemically-defined serum-free medium (“−serum” condition) or 10% FCS as control (“+serum”), and then exposed to serial dilutions of spermidine and spermine. The MTT tests performed 1, 2, and 3 days later confirmed the strong cytotoxic effects of both polyamines at concentrations of ≥100 µM in the presence of 10% FCS (Figure 2a,b). In contrast, under serum-free conditions, spermidine and spermine showed no strong cytotoxic effects at concentrations up to 5 mM, and reduced metabolic activity was only detected at the highest tested concentration of 10 mM (Figure 2a,b). Thus, the avoidance of FCS and the utilization of serum-free medium allows for the study of spermine and spermidine at concentrations almost equivalent to those in semen.

To examine if amine oxidases in FCS are responsible for the conversion of spermine and spermidine into toxic intermediates, we analyzed the effect of the oxidase inhibitor AG. Control experiments showed that AG alone did not exert cytotoxic effects at concentrations of up to 5 mM in TZM-bl cells and primary blood mononuclear cells (PBMC) (Figure A2), confirming previous data obtained with AG in other cells [16,23,25,33]. We then incubated 100% FCS with 0, 50, 500, and 5000 µM of AG for 24 h and supplemented TZM-bl cells with 10% (*v/v*) of these samples, together with spermidine and spermine concentrations of up to 10 mM. Cells were then incubated and MTT assays performed 2 days later. This experiment showed that AG concentrations of 50, 500, and 5000 µM effectively reverted FCS-mediated cytotoxicity of both polyamines (Figure 3a,b). These results confirm that amine oxidases in FCS are responsible for generating toxic polyamine products, as previously suggested [11,16,17,18]. Furthermore, we confirmed that AG functions as a supplement that enables the analysis of high spermine and spermidine concentrations in the presence of FCS. 

We suspected that the use of chemically-defined serum-free medium and/or AG-treated FCS might not only prevent the cytotoxic effects of spermine and spermidine, but also SP. To test this, TZM-bl cells were either incubated with the standard supplement of 10% FCS (FCS), AG-treated FCS (FCS preincubated with 0.5 mM AG), a chemically-defined medium (no FCS), or an AG-treated chemically defined medium (no FCS, 0.05 mM AG). Cells were then exposed to SP concentrations of up 40% (*v*/*v*) and cell viability was determined 2 days later by MTT assay. As shown in Figure 4a, in the presence of FCS, SP was strongly cytotoxic at concentrations of 0.3% and higher, as expected (Figure 1a). When cells were supplemented with AG-treated FCS, SP was less toxic, with still more than 90% viable cells in the presence of up to 10% of SP (Figure 4a). Serum-free conditions also allowed us to analyze SP at concentrations of up to 2.5% (Figure 4a). A combination of both serum-free medium and AG prevented the toxic effects of SP most efficiently, allowing for the analysis of SP concentrations of up to 10 volume% without causing any significant reduction in cell viability (Figure 4a). Similar results were obtained using primary PBMCs instead of immortalized cells (Figure 4b). In the presence of serum, SP caused massive cell death after 3 days, even at concentrations of only 0.2% SP (Figure 4b). The utilization of AG-treated FCS largely prevented toxicity, allowing us to study SP at concentrations of up to 10% in PBMCs. Again, a combination of serum-free medium and AG most effectively reduced SP-induced toxicity (Figure 4b). 

Finally, we evaluated whether the addition of AG and the utilization of serum-free medium may allow for the prevention of SP toxicity in vaginal tissue blocks. For this, tissues derived from 4 donors were dissected into 2 × 2 × 1 mm^3^ blocks, and 10 blocks per donor were cultivated in either a medium containing 10% FCS or a serum-free medium containing 0.05 mM AG. Blocks were then exposed for 3 days to buffer or 10% and 20% SP (*v*/*v*), and cell viability was determined by measuring intracellular ATP levels using a luminescence-based assay (Figure 5 and Figure A3), including a detergent control (0.5% triton-X100). In the presence of FCS, 10% and 20% SP resulted, on average, in an 80% and 90% reduced viability as compared to the buffer control. The addition of AG completely rescued tissue viability in the presence of 10% SP and reduced viability by 25% in the presence of 20% SP. Thus, the supplementation of medium with AG-treated FCS allows for the analysis of SP under conditions not affecting the cell viability of vaginal tissue. 

## 3. Discussion

Our data confirm and expand those of others by showing that (1) spermine and spermidine are cytotoxic in the presence of bovine serum [11,16,17,18,33] and (2) that the amine oxidase inhibitor AG reduces the toxicity of polyamines in the presence of FCS [16,23,25,33]. Moreover, we show that supplementation of FCS containing medium with AG allows researchers to minimize the cytotoxic activity of seminal plasma, suggesting that amine oxidases present in FCS are the responsible factors for the cytotoxic activity of seminal plasma. These enzymes oxidize spermine and spermidine, thereby producing harmful byproducts, such as hydrogen peroxide, ammonia, and reactive aldehydes [18,21,22,25]. Alternatively, the utilization of a serum-free medium supplemented with defined nutrients also allows researchers to minimize the cytotoxic activity of polyamines and/or SP. We furthermore demonstrate that supplementation of an FCS-containing medium with AG or the utilization of chemically-defined serum allows for the investigation of SP in primary PBMC and vaginal tissues blocks. This is of particular importance, as many studies reported an immunosuppressive activity of SP [7,9,20,35,36], which were later questioned and attributed to misinterpretation of data due to the overlooked cytotoxic effects of SP [16,29]. Similarly, the effect of semen and seminal plasma on the infectivity of sexually transmitted viruses is highly controversial, with enhancing or inhibiting effects of SP reported even for the same viral pathogen [12,13,14,37,38,39,40]. Adaption of the protocols described herein will foster future research to clarify the role of SP (or whole semen) on cellular process and viral infections in the recipient’s tissues. 

## 4. Materials and Methods

### 4.1. Reagents 

Spermine and spermidine were purchased from Sigma Aldrich, Darmstadt, Germany (#S3256 and #S2626). Aminoguanidine was obtained from Sigma Aldrich (#109266). The Xvivo-15 growth medium was purchased from Biozym, Hessisch Oldendorf, Germany (#BE02-060F). The DMEM and PRMI 1640 growth mediums were purchased from Life Technologies, Carlsbad, CA, USA (#41965-039 and ##21875-034). Fetal calf serum war purchased from Invitrogen, Waltham, MA, USA (#10270106).

### 4.2. Semen and Seminal Plasma

Semen was provided by the “Kinderwunsch-Zentrum Ulm”, a fertility center in Ulm. The semen of about 50 individual donors was allowed to liquefy for 30 min, then pooled and stored in aliquots at −80 °C. Seminal plasma represents the cell free supernatant of semen, and was prepared by centrifugation at 20,000× *g* for 30 min at 4 °C.

### 4.3. Cell Culture and Primary Tissue

Adherent TZM-bl (HeLa based) cells were obtained through the NIH AIDS Reagent Program, Division of AIDS, NIAID, NIH: TZM-bl cells (Cat#8129) from Dr. John C. Kappes, and Dr. Xiaoyun Wu and were grown in DMEM supplemented with 10% FCS, 2 mM L-glutamine, 100 units/mL penicillin, and 100 µg/mL streptomycin. Cells were checked for mycoplasma contamination on regular basis. Human peripheral blood mononuclear cells (PBMCs) from healthy donors were prepared by Ficoll density centrifugation and activated for 3 days in RPMI medium containing 10% FCS, 100 units/mL penicillin, and 100 µg/mL streptomycin, 1 mM L-glutamine, 10 ng/mL interleukin-2 (IL-2), and 1 µg/mL phytohemagglutinin (PHA). Blocks cut from surgically removed tissue of the cervix and vagina of pelvic organ prolapse patients were cultured in extracellular medium (ECM) consisting of RPMI with 15% FCS, 2 mM L-glutamine, 1 mM sodium pyruvate, non-essential amino acids, 100 units/mL penicillin, 100 µg/mL streptomycin, 100 µg/mL gentamicin, and 25 µg/mL amphotericin B. For experiments in the presence of semen or seminal plasma, growth medium was supplemented with 100 µg/mL gentamicin to prevent bacterial outgrowth. 

### 4.4. Statement

All experiments and methods were performed in accordance with the relevant guidelines and regulations. All experimental protocols were approved by a named institutional/licensing committee. Experiments involving human semen (200/17; 89/16; 351/10) and blood and tissues (88/17) were reviewed and approved by the Institutional Review Board (i.e., the Ethics Committee of Ulm University). Informed consent was obtained from all human subjects. All human-derived samples were anonymized before use. 

### 4.5. Aminoguanidine Treatment of Serum and Medium

For the treatment of FCS with aminoguanidine (AG), FCS was supplemented with 5–0.5 mM AG and then incubated for 24 h at room temperature. The AG-treated FCS was then used as a supplement for the growth medium. For the AG-containing serum-free medium, AG was added at a desired concentration prior to the addition of cells. 

### 4.6. Toxicity Assays for Cells

Adherent cells were seeded at a density of 2 × 10^3^ cells per well in a flat-bottom 96 well plate in growth medium that was changed the next day to 60 µL of indicated medium for treatment with body fluids and to 90 µL for polyamine treatment. Activated PBMCs were washed, resuspended in the indicated medium containing 10 ng/mL interleuckin-2 (IL-2) at a concentration of 1 × 10^6^ cells per ml and then 60 or 90 µL per well were seeded into a 96 well V-bottom plate for body fluids, or polyamine treatment, respectively. Body fluids and compounds were diluted in PBS and 40 µL (semen and seminal plasma), or 10 µL (polyamines) were added to the cells. The viability of adherent cells was determined by a NAD(P)H-dependent colorimetric MTT (3-[4,5-dimethyl-2-thiazolyl]-2,5-dephenyl-2H-tetrazolium bromide) assay according to manufacturer’s instructions. Viability of suspension cells was analyzed using a CellTiter-Glo™ Luminescence Cell Viability Assay (Promega #G7571) according to the manufacturer’s instructions.

### 4.7. Toxicity Assay for Primary Tissues

Vaginal tissue was cut into 2 × 2 × 1 mm^3^ blocks as described before [41,42]. Ten blocks per treatment were then treated with 10% or 20% seminal plasma (or buffer) in the presence of chemically defined, serum-free medium (X-vivo15) supplemented with 0.005 mM aminoguanidine or in the presence of ECM growth medium supplemented with 10% FCS. After 3 days, the individual blocks were washed twice with PBS and analyzed using a CellTiter-Glo™ viability assay. 

### 4.8. Statistical Analyses

Descriptive data are represented by arithmetic mean and standard error of the mean (SEM). The *p*-values (Figure 5) are calculated based on linear contrast hypotheses following two-way ANOVA, where *p* < 0.05 was considered to be statistically significant. All analyses were conducted in GraphPad Prism (version 9.3.1) and SAS (version 9.4).

## 5. Conclusions

Working with relevant concentrations of polyamines and semen in cell culture or tissues has been problematic due to their well-known cytotoxic effects. Using the amine oxidase inhibitor AG as a cell culture supplement and using serum-free conditions prevents the formation of toxic seminal polyamine metabolites and, thus, enables the investigation of semen and polyamines in realistic physiological conditions. Application of this new method will help to shed light on the effect of semen or seminal plasma on some long-lasting scientific questions, e.g., in the field of sexual viral transmission and cellular processes during fertilization.

## Figures and Tables

**Figure 1 ijms-23-08563-f001:**
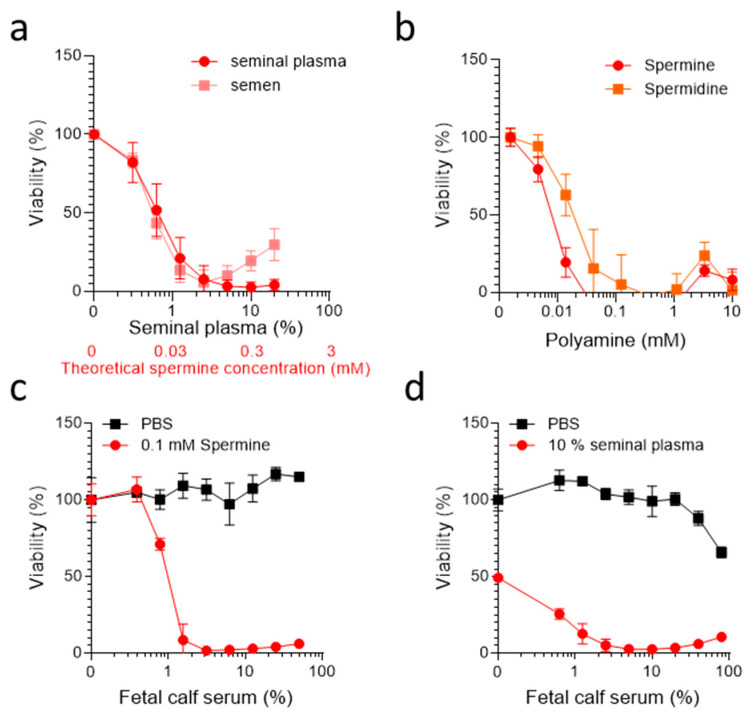
**Semen, seminal plasma, spermine, and spermidine are cytotoxic in the presence of FCS.** (**a**) The TZM-bl cells (2 × 10^3^) were treated with serially diluted pooled seminal plasma or semen in the presence of 10% FCS. (**b**) The TZM-bl (2 × 10^3^) cells were treated with serially diluted synthetic spermine or spermidine in the presence of 10% FCS. (**c**,**d**) The TZM-bl cells (2 × 10^3^) were exposed to 0.1 mM spermine or PBS (**c**), or 10% pooled SP and PBS (**d**) in the presence of DMEM supplemented with indicated concentrations of FCS. (**a**–**d**) The viability was determined 2 days later by MTT assay. Data shown are average % values derived from three individual experiments (**a**,**b**) or one representative experiment (**c**,**d**) performed in biological triplicates ± SD.

**Figure 2 ijms-23-08563-f002:**
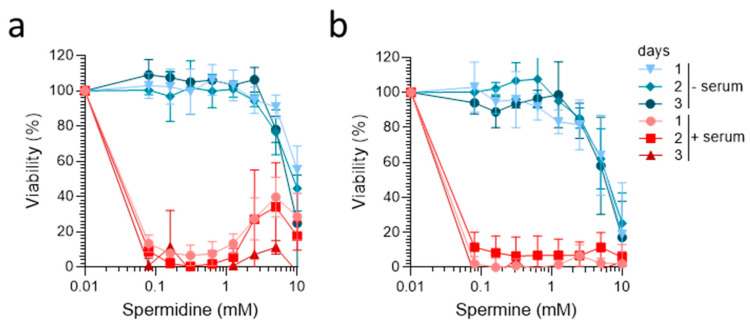
**Spermine and spermidine are not cytotoxic under serum-free conditions.** The TZM-bl cells (2 × 10^3^) were incubated with the indicated concentrations of (**a**) spermidine and (**b**) spermine in the presence of medium with 10% FCS (+serum) or chemically defined serum-free medium (−serum). Cell viability was determined after incubation for 1, 2, and 3 days using MTT assay. Data shown are average % values derived from three individual experiments performed in biological triplicates ± SD.

**Figure 3 ijms-23-08563-f003:**
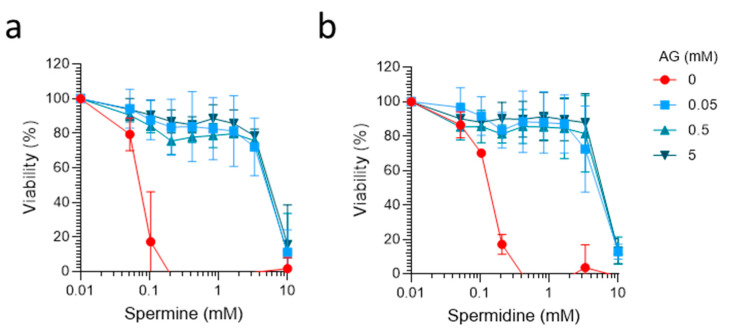
**The diamine oxidase inhibitor aminoguanidine prevents FCS-mediated cytotoxic effects of spermine and spermidine.** Here, FCS (100%) was preincubated with the indicated concentrations of aminoguanidine (AG) for 24 h. Tenfold dilutions of these samples were then used as cell culture supplement for TZM-bl cells, which were incubated with spermine (**a**) and spermidine (**b**) for 2 days. Viability was determined by MTT assay. Data shown are average % values derived from 3 individual experiments performed in biological triplicates ± SD.

**Figure 4 ijms-23-08563-f004:**
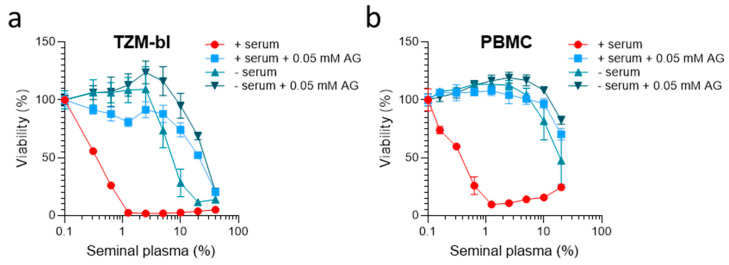
**Aminoguanidine combined with serum-free medium prevents seminal plasma-derived cytotoxic effects.** (**a**,**b**) The TZM-bl (2 × 10^3^) cells (**a**) or PBMCs (1 × 10^5^) (**b**) were treated with the indicated amounts of SP in the presence of growth medium supplemented with 10% FCS, growth medium supplemented with 10% AG-treated FCS (pretreatment with 0.5 mM AG), a chemically-defined, serum-free medium, or a chemically-defined, serum-free medium supplemented with 0.05 mM AG. Viability was determined after 2 days by MTT assay (**a**) or 3 days using CellTiter-Glo™ assay (**b**). Data shown are average % values derived from one representative experiment performed in biological triplicates ± SD.

**Figure 5 ijms-23-08563-f005:**
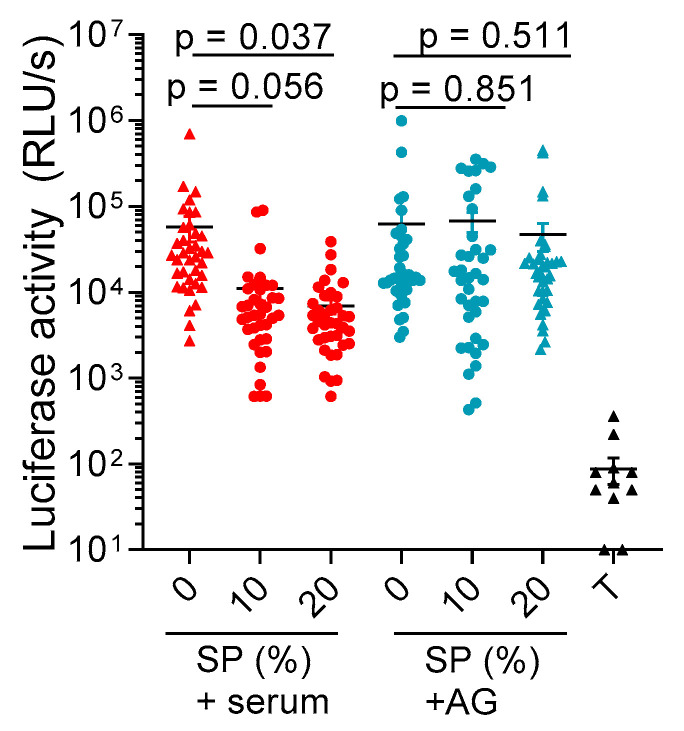
**AG prevents SP-induced cytotoxic effects in vaginal tissue blocks.** Here, 40 (2 × 2 × 1 mm^3^) vaginal tissue blocks derived from 4 individual donors were incubated with buffer, 10%, or 20% SP, in the presence of a chemically-defined, serum-free medium supplemented with 0.05 mM AG or in the presence of normal growth medium supplemented with 10% FCS. After 3 days, the individual blocks were washed twice with PBS and intracellular ATP levels were analyzed using a CellTiter-Glo™ viability assay. Additionally, 0.5% triton-X100 (T) was used as a positive toxicity control. Black lines indicate arithmetic means ± SEM, while RLU/s is defined as relative light units per second. The *p*-values are obtained by means of linear contrast hypothesis tests following two-way ANOVA.

## Data Availability

Primary data are available upon reasonable request.

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
