# Peer review of "Utilization of Aminoguanidine Prevents Cytotoxic Effects of Semen"

_ijms, 2022, doi:10.3390/ijms23158563_

Round 1

Reviewer 1 Report

The manuscript (ijms-1819759) entitled “Utilization of aminoguanidine prevents cytotoxic effect of semen” indicates that aminoguanidine inhibits amine oxidases in semen, and decreases the degradation of spermidine and spermine.  As the results, it is expected that the level of a toxic compound acrolein, produced from spermidine and spermine by amine oxidase, is decreased by aminoguanidine.

Although the effect of aminoguanidine on cell viability was clearly shown, the level of acrolein produced is not shown. So, please indicate how the level of acrolein is changed by aminoguanidine.

Minor points

1.     Fig. 1A, L-Ornithin L-Ornithine

2.     Fig. 5, triton triton X-100 (?)

Author Response

Response to reviewer 1

  1. Although the effect of aminoguanidine on cell viability was clearly shown, the level of acrolein produced is not shown. So, please indicate how the level of acrolein is changed by aminoguanidine.

Response: We planned to quantify acrolein levels generated by co-incubation of spermine and spermine-containing media with FCS in presence/absence of AG using DBB-derivatization and quantitative determination by HPTLC or HPLC [1]. However, both acrolein standards and the derivatization reagent have unexpectedly long shipping times, making it impossible to perform these experiments in a timely manner. Further, Aminoguanidine is a well described potent inhibitor for amino oxidases (AOs) [2]. It has previously been demonstrated, that aminoguanidine can prevent the formation of H202, which together with acrolein is generated during amine oxidation by AOs [3]. Therefore, it is highly expected that aminoguanidine also decreases the generation of acrolein. The toxicity results obtained in our study support this assumption.

[1]          T. Imazato et al., “Determination of acrolein in serum by high-performance liquid chromatography with fluorescence detection after pre-column fluorogenic derivatization using 1,2-diamino-4,5-dimethoxybenzene,” Biomed. Chromatogr., vol. 29, no. 9, pp. 1304–1308, Sep. 2015, doi: 10.1002/bmc.3422.

[2]          H. Tamura, K. Horiike, H. Fukuda, and T. Watanabe, “Kinetic studies on the inhibition mechanism of diamine oxidase from porcine kidney by aminoguanidine.,” J. Biochem., vol. 105, no. 2, pp. 299–306, Feb. 1989, doi: 10.1093/oxfordjournals.jbchem.a122657.

[3]          L. Wang et al., “Oxidative degradation of polyamines by serum supplement causes cytotoxicity on cultured cells.,” Sci. Rep., vol. 8, no. 1, p. 10384, Jul. 2018, doi: 10.1038/s41598-018-28648-8.

  1. Fig. 1A, L-Ornithin → L-Ornithine

Response: corrected

  1. Fig 5, triton → triton X-100 (?)

Response: We thank the reviewer for pointing this out. We corrected it in the figure 5 legend and in the text on page 6.

Reviewer 2 Report

The research conducted is of high interest and the experimental design appropriate. The results obtained are accurately presented and the discussion focus on the most relevant aspects of the research conducted. 

In spite of the high quality of the manuscript there are two aspects that authors must addres:

1) Add a Conclusions section highlighting the relevance of their study.

2) Review the Reference list and adapt it to the format of the journal.

Author Response

Response to reviewer 2

In spite of the high quality of the manuscript there are two aspects that authors must address:

  1. Add a Conclusions section highlighting the relevance of their study.

Response: We thank the reviewer for the helpful comment. A conclusion section has been added.

  1. Review the Reference list and adapt it to the format of the journal

Response: We changed the style of the references according to the journals format.